# Hypervitaminosis A in type 2 diabetes and its relation with renal function and cardiovascular complications

**Anna Toffalini**[1]**, Nicolò Vigolo**[1]**, Elena Sani**[1]**, Elisa Paviati**[2]**, Matteo Gelati**[2]**, Elisa Danese**[2]**, Giacomo Zoppini**[iD][1]*

**1** Endocrinology, Diabetes and Metabolism, Department of Medicine, University and Hospital Trust of Verona, Verona, Italy, **2** Section of Clinical Biochemistry, Department of Engineering for Innovation Medicine, University and Hospital Trust of Verona, Verona, Italy

* giacomo.zoppini@univr.it

## Abstract

Vitamin A is a fundamental micronutrient in the diet. In recent years, hypervitaminosis A has been described in subjects, mainly children, with renal dysfunction. The aim of the present study was to estimate the prevalence of hypervitaminosis A in adult patients affected by type 2 diabetes and to evaluate the associated factors. This cross-sectional observational study included 200 ambulatory patients affected by type 2 diabetes regularly attending the Diabetes Clinic of Verona (Italy). Eligible patients were consecutively enrolled. Vitamin A was measured by reverse phase high performance liquid chromatography with UV detection. The normal range of values for vitamin A concentrations was 1.04–2.79 µmol/L. All other laboratory parameters were measured by standard methods. A multivariate logistic regression model was applied to study the factors independently associated with the hypervitaminosis A. Hypervitaminosis A was detected in 29% of our cohort of patients. Its prevalence was significantly higher in patients with cardiovascular complications (CVD). Nevertheless, when patients with CVD were stratified according to renal function, the latter emerged as the main factor associated with hypervitaminosis A. The multivariate analysis confirmed that estimated glomerular filtration rate was an independent predictor of hypervitaminosis A (0.97 CI 95% 0.96–0.99, p = 0.008). The main alteration of vitamin A in ambulatory patients affected by type 2 diabetes is hypervitaminosis. Renal function is strongly associated with hypervitaminosis A.

## Introduction

Vitamin A is a fundamental micronutrient in the diet [1]. Vitamin A is a group of fat-soluble compounds derived from both vegetables and animal foods that present a common unsaturated isoprenoid chain structure [2]. The predominant form of retinoids is retinol [3]. All these compounds are liposoluble and accumulate in the

**Data availability statement:** The data analyzed in the current study due to policy of the Research Ethics Committee of the Hospital Trust of Verona. This precaution is taken as there is a potential that participants can be identified based on their individual data. However, the data are available from the University of Verona (Casella Endocrinologia Diabetologia Malattie Metabolismo endocrinologia.metabolismo@aovr.veneto.it) upon a reasonable request.

**Funding:** The author(s) received no specific funding for this work.

**Competing interests:** The authors have declared that no competing interests exit.

body, especially in the liver and adipose tissue [3]. The kidneys are implicated in the metabolism and excretion of vitamin A [4].

In healthy subjects, dietary vitamin A is processed to retinol, deposited in the liver and transported by retinol-binding-protein (RBP4) and transthyretin (TTR) to the cells of the body [2,3]. Retinol is oxidized to its active form, retinoic acid (RA) [5]. Impaired renal function has been associated with high circulating levels of retinol, possibly due to a combination of decreased glomerular filtration rate of the retinol-RBP4 complex, reduced conversion of retinol to retinoid acid, and an accumulation of RBP4 [6].

In recent years, hypervitaminosis A, not only due to supplements, has appeared in literature more frequently. Since vitamin A excess may have serious clinical consequences [7], it is of interest to have a better understanding of this entity in different clinical situations. In children with chronic renal disease (CKD), hypervitaminosis A was reported as early as stage G2 of CKD [8,9]. Previous studies in adults [10–12] and children [13–15] on dialysis have found hypervitaminosis A. Therefore, it is important to recognize conditions associated with hypervitaminosis A in order to avoid excessive vitamin A intake or supplementation.

Type 2 Diabetes (T2D) is a common chronic disease that can lead to cardiac and renal complications [16]. Data on hypervitaminosis A are lacking in adults with T2D; therefore the aims of the present study were to estimate the prevalence of hypervitaminosis A and its associations with cardiovascular and kidney complications in subjects affected by T2D.

## Materials and methods

The present cross-sectional observational study comprised 200 patients with T2D attending the Diabetes Clinic of the University of Verona, who were consecutively enrolled. Inclusion criteria were age between 18 and 80 years, both sexes and T2D diagnosed with standard criteria at least 3 months before enrollment. Exclusion criteria included a history of pernicious anemia, autoimmune gastritis, pregnancy and any vitamin supplementation within 6 months before the study. All subjects enrolled in the study should have been free of vitamin supplements in the six months preceding the determination of vitamin A in order to eliminate interferences.

The study was approved by the local ethics committee of the Hospital Trust of Verona (n°3853CESC), and informed written consent was obtained from each participant. The study was conducted in accordance with the Declaration of Helsinki. The recruitment period was from 1/09/2022 and 6/04/2023.

After an overnight fast (8–12 hours), venous blood samples were collected in the morning. Serum creatinine (Roche enzymatic method) and other biochemical blood measurements: glycemia, lipid profile and aspartate aminotransferase (using standard laboratory procedures, DAX 96; Bayer Diagnostics, Milan, Italy) were determined. Low-density lipoprotein-cholesterol was calculated using the Friedewald's equation [17]. Hemoglobin A1c (HbA1c) was measured by an automated high-performance liquid chromatography analyzer (HA-8140; Menarini Diagnostics, Florence, Italy). The glomerular filtration rate (GFR) was estimated by the CKD Epidemiology Collaboration (CKD-EPI) equation [18]. Albuminuria was measured by an

immuno-nephelometric method on a morning spot urine sample and expressed as the albumin-to-creatinine ratio. Patient under study during the scheduled blood test for the medical control underwent a further blood test for the measurement of vitamin A. Vitamin A was measured by reverse phase high performance liquid chromatography (HPLC) with UV detection (HPLC Prominence, Shimadzu). The normal range of values of vitamin A was 1.04–2.79 μmol/L.

Duration of diabetes was computed from the date of diagnosis of diabetes. Height and weight were measured using a calibrated stadiometer and balance-beam scale, respectively. Body mass index (BMI) was calculated by dividing weight in kilograms by height in meters squared. A physician measured blood pressure with a mercury sphygmomanometer after patients had been seated quietly for at least 5 minutes. Patients were considered to have hypertension if their blood pressure was ≥ 140/90 mmHg or if they were taking any anti-hypertensive drugs. In all participants, the presence of micro-vascular diabetic complications was recorded. A single ophthalmologist diagnosed diabetic retinopathy using fundoscopy after pupillary dilation according to a clinical disease severity scale (no retinopathy, non-proliferative, proliferative or laser-treated retinopathy); the presence of proliferative retinopathy was confirmed by fundus fluorescein angiography. Nephropathy was defined as the presence of eGFR < 60 ml/min/1.73 m$^2$ and/or abnormal albuminuria (i.e., an albumin-to-creatinine ratio ≥ 30 mg/g creatinine). A confirmed history of myocardial infarction, angina, coronary revascularization, stroke, transitory ischemic attack, carotid revascularization, non-traumatic amputation, gangrene and/or lower limb revascularization was considered a valid proxy for prior clinical cardiovascular disease (CVD). Ultrasonography scanning of common and internal carotid arteries was performed as previously described (Esaote Wall Track System, Esaote S.p.A., Genova, Italy) and a cut-off of 60% was used to define a significant arterial stenosis. Ultrasonography scanning of lower limb arteries was performed and any detected stenosis or moderate-to-severe reduction of blood flow at proximal and/or distal level was considered as a marker of peripheral artery disease.

## Statistical analysis

Data were summarized as mean ± SD for continuous variables and absolute value or percentages for categorical variables. Differences in clinical/ biochemical characteristics were tested with the Student's t-test for normally distributed variables and the Mann–Whitney test for non-normally distributed variables. ANOVA were used for variables with three or more categories. The χ2 test was used for categorical variables to study differences in proportions or percentages between the two groups. In order to better evaluate the relation of vitamin A levels and atherosclerotic burden, we created a categorical variable (CVD) with 0 equal to the absence and 1 equal to the presence of ischemic heart diseases or peripheral artery disease or cerebrovascular disease or previous carotid thromboarterectomy.

A logistic regression model was performed to investigate the independent predictors of hypervitaminosis A. Covariates for this multivariate regression model were chosen as potential confounders based on their biological plausibility. A p value of less than 0.05 was considered statistically significant. The Analyses were carried out with SPSS Statistics for Data Analysis v.20.0.1.1.

## Results

Twenty nine percent of patients were found to have hypervitaminosis A, whereas only 2 subjects had hypovitaminosis A (1%). The main clinical and anthropometric characteristics of patients are presented in Table 1, stratified by hypervitaminosis A. Patients in the hypervitaminosis A group were significantly older with a longer duration of diabetes. The BMI tended to be lower, whereas the eGFR was significantly lower (72.6 ± 21.5 vs 83.5 ± 18.8, p < .001) in the hypervitaminosis A group. The sex distribution was substantially the same, 22.4% women and 32.2% men with a p value of 0.144. The prevalence of obesity was significantly lower in the hypervitaminosis A group (18.8% obesity vs 34.4% no obesity, p = 0.022). A significant linear inverse correlation was found between vitamin A concentrations and eGFR, r = 0.302 and p < 0.001.

We further evaluated the distribution of cardiovascular diseases and arterial hypertension with hypervitaminosis A. As shown in Fig 1, the prevalence of hypervitaminosis A was significantly higher in both patients with cardiovascular diseases

**Table 1. Clinical characteristics of ambulatory patients with type 2 diabetes in relation to hypervitaminosis A (HVA).**

|  | No HVA (n = 142) | HVA (n = 58) | p |
|---|---|---|---|
| Vitamin A, µmol/l | 2.12 | 3.14 | NA |
| Age, yr | 65.8 ± 9.1 | 68.8 ± 10.0 | .029 |
| Duration of diabetes, yr | 11.4 ± 7.8 | 14.0 ± 8.7 | .049 |
| BMI, Kg/m2 | 28.5 ± 4.8 | 27.2 ± 4.0 | .063 |
| Systolic blood pressure, mmHg | 132.9 ± 18.4 | 136.9 ± 18.3 | .258 |
| Diastolic blood pressure, mmHg | 80.5 ± 10.6 | 78.9 ± 10.5 | .429 |
| Glycated hemoglobin, mmol/mol Hb | 52.8 ± 11.5 | 54.1 ± 11.0 | .437 |
| Total cholesterol, mg/dl | 145.7 ± 35.2 | 142.1 ± 35.0 | .329 |
| HDL-cholesterol, mg/dl | 49.1 ± 13.2 | 49.8 ± 20.7 | .666 |
| LDL-cholestrol, mg/dl | 72.6 ± 28.9 | 67.8 ± 30.0 | .303 |
| Triglycerides, mg/dl | 119.5 ± 99.5 | 124.3 ± 56.9 | .345° |
| AST, U/L | 24.9 ± 13.4 | 24.6 ± 11.2 | .866 |
| ACR, mg/gr | 59.9 ± 218.4 | 184.6 ± 468.9 | .060° |
| eGFR, ml/min 1.73 m2 | 83.5 ± 18.8 | 72.6 ± 21.5 | <.001 |

° Test Mann-Whitney, otherwise t-test. Data are presented as mean±standard deviation. BMI, body mass index; HDL, high density lipoprotein; LDL, low density lipoprotein; AST, aspartate transaminase; ACR, albumin creatinin ratio; eGFR, estimated glomerular filtration rate; NA, not applicable.

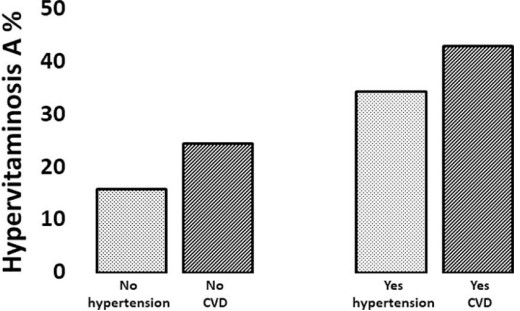

**Fig 1. The histograms represent the prevalence of hypervitaminosis A according to hypertension and cardiovascular diseases.** The $X^2$ for hypertension was p = 0.006 and for CVD was p = 0.013.

and those with arterial hypertension. Since the eGFR was strongly correlated with vitamin A, we stratified subjects with and without CVD and hypertension according to kidney function. Fig 2 shows that when the kidney function is reduced (eGFR < 60 ml/min 1.73m$^2$), the concentrations of vitamin A were higher (p for trend = 0.004). The proportion of patients with hypervitaminosis A compared to those without was significantly higher ($X^2$, p = .004) both in patients with no CVD/yes CKD (15.5% vs 5.7%) and in patients with yesCVD/yesCKD (12.1% vs 7.8%).

Moreover, the vitamin A level increased throughout the categories of CKD; G1 = 2.26 ± 0.48; G2 = 2.40 ± 0.68; G3a = 2.72 ± 0.71; G3b 2.56 ± 0.61 and G4 (only 3 subjects) 3.45 ± 0.79, p for trend <0.001. The increasing level of vitamin A was significant starting from the CKD3a category (post hoc test of Scheffè).

Table 2 shows the multivariate logistic analysis with hypervitaminosis A as the dependent variable. eGFR was significantly associated with hypervitaminosis A (OR 0.97, CI 95%, 0.96–0.99, p = 0.008). A significant inverse association with obesity (OR 0.43, CI95% 0.20–0.90, p = 0.025) was also found.

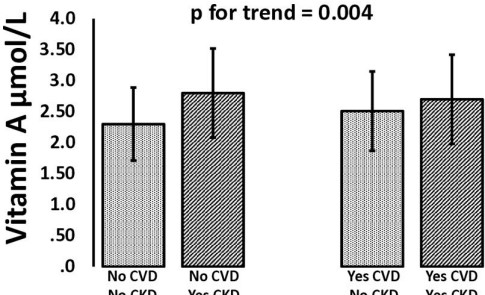

**Fig 2. The histograms show the concentrations of vitamin A in subjects categorize for the presence and absence of CVD with or without CKD.** The p for trend was equal to 0.002.

**Table 2. Multivariate logistic regression analysis with Hypervitaminosis A as dependent variable.**

| variables | | |
|---|---|---|
| | OR (CI-95%) | p |
| Age, yrs | 1.01 (0.96-1.06) | .772 |
| Obesity, Yes | 0.43 (0.20-0.90) | .025 |
| eGFR, ml/min | 0.97 (0.96-0.99) | .008 |
| HbA1c, mmol/mol | 1.02 (0.99-1.05) | .244 |
| Sex, M | 1.61 (0.79-3.30) | .193 |

*Atherosclerotic diseases: 0 equal to the absence and 1 equal to the presence of ischemic heart diseases or peripheral artery disease or cerebrovascular disease or previous carotid thromboarterectomy.

## Discussion

In this study, the most frequent alteration of vitamin A concentration in ambulatory patients with T2D was hypervitaminosis. This condition was present in 29% of subjects. Another main result of the study was the strong relationship of vitamin A with kidney function in these patients. Notably, vitamin A concentration began to increase from stage CKD G3a. The correlation between vitamin A levels and eGFR was confirmed in the multivariate logistic analysis. The higher prevalence of CVD and arterial hypertension in the hypervitaminosis A group seemed to be mediated by the reduced kidney function. It is important to emphasize that care was taken to exclude subjects who used vitamin supplements in the six months before the study. This step was critical in order to eliminate supplements as possible cause of hypervitaminosis A.

An older study measured vitamin A concentration in type 1 (n = 35) and type 2 diabetes (n = 35) compared to controls (n = 30). This study found that vitamin A was decreased in type 1 and increased in type 2 diabetes [19]. The correlation with kidney function was not explored [19]. In more recent years, hypervitaminosis A has become an object of investigation.

In previous studies hypervitaminosis A was reported in children with renal dysfunction and in both children and adults on dialysis [8–15]. In adults, all stages of CKD compared to healthy controls were associated with higher levels of vitamin A [6]. The higher levels of this vitamin were observed in patients with CKD stages G3-G5 and in dialyzed patients [6]. Similarly, in our study vitamin A level began to increase from the CKD G3a stage.

To the best of our knowledge, this study is the first description of the presence of hypervitaminosis A in adults with T2D and its relationship with kidney function. The role of vitamin A in CVD development is unclear; nevertheless, some studies have shown an association with cardiovascular outcomes [20]. In our study, hypervitaminosis A was associated with cardiovascular complications in type 2 diabetes. However, after stratification for renal function, vitamin A levels were mainly

affected by eGFR and not by the presence of CVD. A similar impact of renal dysfunction, in accordance with our results, was shown in another study that measured RBP4 in adult patients with type 2 diabetes [21].

The results of our study may have clinical implications. First, as high levels of vitamin A have been associated with several adverse effects [22–27], it is important to measure vitamin A levels in patients with type 2 diabetes before starting vitamin A supplementation and to carefully monitor the concentrations [28], especially in patients with reduced renal function. Second, complications related to high levels of vitamin A including decreasing renal function [29,30] may benefit from reducing increased levels of vitamin A. Therefore, we can hypothesize that reducing vitamin A levels may have clinical benefits. Third, vitamin A levels may be merely an initial marker of declining renal function, as reported for RBP4 [31].

The mechanisms underlying this association are still matter of research. Kidneys are implicated in the regulation of vitamin A and its carrier protein retinol binding protein-4 metabolism [32], but the liver is the main organ for the clearance of vitamin A [33].

The mechanism proposed for the increased level of vitamin A is a reduced renal clearance [8,9]. Nevertheless, an increased hepatic release of retinol can be a concomitant mechanism [34]. The latter effect can be caused by altered signaling between the kidney and liver in CKD [6]. The effect of kidney disease on hepatic function has been well described for hepatic cytochrome P450 enzymes [35]. Moreover, a recent and interesting study showed that hypertensive nephropathy status may impact on the expression of drug-metabolizing enzymes and transporters in the liver and kidney, thus influencing the metabolisms of substrate drugs in vivo [36].

Important weaknesses of our study are the absence of a control group, and the cross-sectional design, which limits causal inferences, and allows for only associations, not causal relationships. The sample size was difficult to determine from previous studies, as they were carried out in children and patients on dialysis. However, our sample of 200 subjects was larger than all the samples in the other studies; thus, we believe that our finding may be a reasonable estimate of hypervitaminosis A. The strengths of the study are the number of participants, the care taken to exclude patients who took vitamin supplements, and the completeness of the database.

## Conclusions

In conclusion, in patients with type 2 diabetes, hypervitaminosis A seems to be a frequent finding. Hypervitaminosis A is related to renal function, therefore future studies on the post-translational modifications in kidney disease may help clarify our findings.

## Author contributions

**Conceptualization:** Elisa Danese, Giacomo Zoppini.

**Data curation:** Anna Toffalini, Nicolò Vigolo, Elena Sani, Elisa Paviati, Matteo Gelati.

**Formal analysis:** Giacomo Zoppini.

**Methodology:** Elisa Paviati, Matteo Gelati, Elisa Danese.

**Supervision:** Elisa Danese, Giacomo Zoppini.

**Writing – original draft:** Giacomo Zoppini.

**Writing – review & editing:** Elisa Danese, Giacomo Zoppini.

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
