## [Decision Letter · Decision Letter 0]

18 Mar 2025

PONE-D-24-58961HYPERVITAMINOSIS A IN TYPE 2 DIABETES AND ITS RELAZION WITH RENAL FUNCTION AND CARDIOVASCULAR COMPLICATIONS.PLOS ONE

Dear Dr. Zoppini,

Thank you for submitting your manuscript to PLOS ONE. After careful consideration, we feel that it has merit but does not fully meet PLOS ONE’s publication criteria as it currently stands. Therefore, we invite you to submit a revised version of the manuscript that addresses the points raised during the review process.

**ACADEMIC EDITOR: Kindly revise and resubmit as per the reviewer's comments**

We look forward to receiving your revised manuscript.

Kind regards,

Parasuraman Pavadai

Academic Editor

PLOS ONE

[None].

4. In the online submission form, you indicated that [The data of this study are available from the corresponding author upon reasonable request. The data cannot be accessed publicly due to specific limitations that could compromise the confidentiality and privacy of the participants.].

Additional Editor Comments:

Based on the reviewer's comment kindly revise and resubmit.

Reviewers' comments:

Reviewer's Responses to Questions

**Comments to the Author**

1. Is the manuscript technically sound, and do the data support the conclusions?

Reviewer #1: Partly

Reviewer #2: Partly

Reviewer #3: Partly

2. Has the statistical analysis been performed appropriately and rigorously? 

Reviewer #1: Yes

Reviewer #2: Yes

Reviewer #3: No

3. Have the authors made all data underlying the findings in their manuscript fully available?

Reviewer #1: Yes

Reviewer #2: Yes

Reviewer #3: No

4. Is the manuscript presented in an intelligible fashion and written in standard English?

Reviewer #1: No

Reviewer #2: Yes

Reviewer #3: No

5. Review Comments to the Author

Reviewer #1: Abstract

Under “Research Design and Method”, please provide statements on the specific study site, including the name of the study site and the sampling technique adopted for this study.

Under “Results” provide figures to support the statement “The HVA prevalence was significantly higher in patients with cardiovascular complications (CVD). Nevertheless, when patients with CVD were stratified according to renal function, the latter emerged as the main factor associated with HVA”. For example what is the proportion and the p-value? These are necessary to appreciate the outcome of this study.

Under “Conclusions”, the statement “The main alteration of vitamin A in ambulatory patients affected by type 2 diabetes is hypervitaminosis” should be rephrased for clarity.

Main Text

Introduction

The last statement should be moved to the methodology subsection. It is most appropriate to be included in the statements of inclusion and exclusion criteria.

Under “Research Design and Method”, by describing the study site as “our diabetic clinic” does not help international readership. Please state the exact name for the study site.

State the specific duration for the overnight fast.

What constitutes "other biochemical measurements"?

The statement on height and weight measurement should come before body mass index calculation.

Before Statistical Analysis subsection, the authors should provide statements on the sample size determination.

Concerning these statements under “Results”, Vitamin A was determined in 200 ambulatory patients affected by type 2 diabetes. Patients who took vitamin supplements within the six months before the study were excluded. The cut-off levels of vitamin A were ≤ 1.04 and > 2.79 μmol/L for either hypovitaminosis or hypervitaminosis, respectively. These statements cannot be the narratives for the result outcomes. They are better suited for the subsection on methodology.

The authors found that “The sex distribution of HVA was substantially the same, 22.4% women and 32.2% men with a p value of 0.144.” What sampling technique did the authors adopt in arriving at this outcome for the sex distribution?

The authors also suggested that “the levels of VA increased throughout the categories of CKD; G1= 2.26±0.48; G2= 2.40±0.68; G3a= 2.72±0.71; G3b 2.56±0.61 and G4 (only 3 subjects) 3.45±0.79, p for trend <0.001. The increasing level of vitamin A was significant starting from the CKD3a category (post hoc test of Scheffè). These statements of result claimed by the authors in Figure 2 do not exist in the said Figure 2. So it was difficult to verify the outcome.

In the narrative for Table 2, the authors should report the odd ratio, confidence interval and p-values for the association between obesity and HVA.

The subsection “Conclusions” should be rendered “Discussions”

Under “Conclusions”, The statement “Notably, vitamin A concentration began to increase from stage CKD G3a is not supported by the said "Figure 2 ". Please clarify.

Reviewer #2: I appreciate the opportunity to review this manuscript and contribute to the scholarly process. The study addresses an important yet underexplored aspect of metabolic health, and I look forward to providing constructive feedback.

General Comments:

1. Please add line numbers to facilitate clear and precise communication during the review process.

2. Include a title for the Background or Introduction section after the abstract for better organization.

3. Consider structuring the final section as Discussion and Conclusion rather than just Conclusion to ensure a comprehensive synthesis of findings.

Specific comments

1. The term "RELAZION" in the title is unclear. Could you clarify its intended meaning? If you meant "relation" or "relationship," I recommend revising the title for clarity and accuracy.

2. What are the implications of hypervitaminosis A on diabetes treatment?

3. Does it pose a real concern for diabetes outcomes, or could it potentially influence treatment efficacy?

4. Are there any established guidelines recognizing hypervitaminosis A as a complication, a factor influencing disease progression, or a determinant of treatment outcomes?

5. How does this study contribute to the existing body of knowledge, and what is its practical significance for clinicians managing diabetes?

6. Are you suggesting that hypervitaminosis A could serve as an indicator of early-stage chronic kidney disease (CKD) in diabetic patients?

7. If so, what are the potential mechanisms by which hypervitaminosis A could contribute to or reflect kidney dysfunction in this context?

8. What are the clinical implications of these findings for early diagnosis and management of CKD in diabetic individuals?

Reviewer #3: Title language should be in english - relation and not relazion

Introduction -

Heading is missing

Vitamin A shoild not be abbreviated

Retinoic and not retinoid acid

Introduction doesnt emphasize the role of Vitamin A in Type 2 diabetes. Direct implication of Vit A in type 2 diabetes was nto mentioned.

Research design and method

Inclusion criteria reason out the age range and sex. Was there any grouping done based on age or sex. If not why wasit not done. Did the patients used insulin supplement? Were they included or excluded. Does exclusion criteria inludes any terminal or old age related illness. What were the biochemical blood measurements done. Not reported in the study. Only creatine was done. Renal function test was not carried out. Severity scale was not mentioned in the research design. Grouping based on severity of the diseases was not done.The sentences should not use the word WE or I. Past tense should be used. is to be corrected to was.

"The main clinical and anthropometric characteristics" - where are the anthrometric characteristics

Correlation between Cardiovascular disease (CVD) and Vit A was not done. What are evaluations done withrespect cardiovascular.

Conclusion

Tense correction in sentences. Should be in one tense. "We" should not be used.

Comparison with other studies which were carried out between Vita A and CVD or Vit A and Renal should be added. Revieing past literature with present study should be carried out. The conclusion doesnt provide information on statistical significane or correlation.

Review. Significant defiency in the study design and methodology doesnt substanstiate the role of Vit A in CVD and Renal function.

6. PLOS authors have the option to publish the peer review history of their article (what does this mean? ). If published, this will include your full peer review and any attached files.

**Do you want your identity to be public for this peer review?** For information about this choice, including consent withdrawal, please see our Privacy Policy .

Reviewer #1: **Yes: ** Dr. Sylvester Yao Lokpo

Reviewer #2: No

Reviewer #3: No

---

## [Author Response · Author response to Decision Letter 1]

1 Apr 2025

Review Comments to the Author

A. We reviewed the manuscript for style requirements.

[None].

A. We include this information in the cover letter.

A. Our data are available on request. See question 4.

4. In the online submission form, you indicated that [The data of this study are available from the corresponding author upon reasonable request. The data cannot be accessed publicly due to specific limitations that could compromise the confidentiality and privacy of the participants.].

A. The data analyzed in the current study due to policy of the Research Ethics Committee of the Hospital Trust of Verona. This precaution is taken as there is a potential that participants can be identified based on their individual data. However, the data are available from the University of Verona (Casella Endocrinologia Diabetologia Malattie Metabolismo endocrinologia.metabolismo@aovr.veneto.it) upon a reasonable request.

We reported the above sentence in the submission section.

A.Done

Reviewer #1: Abstract

Under “Research Design and Method”, please provide statements on the specific study site, including the name of the study site and the sampling technique adopted for this study.

A. We add the statements.

Under “Results” provide figures to support the statement “The HVA prevalence was significantly higher in patients with cardiovascular complications (CVD). Nevertheless, when patients with CVD were stratified according to renal function, the latter emerged as the main factor associated with HVA”. For example what is the proportion and the p-value? These are necessary to appreciate the outcome of this study.

A. Thanks, we analyzed the proportion and added the following sentence in the results section “The proportion of patients with hypervitaminosis A compared to those without was significantly higher (Χ2, p=.004) both in patients with noCVD/yes CKD (15.5% vs 5.7%) and in patients with yesCVD/yesCKD (12.1% vs 7.8%).”

The figure was very similar to the figure2 already present in the manuscript.

Under “Conclusions”, the statement “The main alteration of vitamin A in ambulatory patients affected by type 2 diabetes is hypervitaminosis” should be rephrased for clarity.

A. We rephrased the statement.

Main Text

Introduction

The last statement should be moved to the methodology subsection. It is most appropriate to be included in the statements of inclusion and exclusion criteria.

A. The statement was moved to methodology subsection.

Under “Research Design and Method”, by describing the study site as “our diabetic clinic” does not help international readership. Please state the exact name for the study site.

A. the study site was specified.

State the specific duration for the overnight fast.

A. We included the duration of overnight fast.

What constitutes "other biochemical measurements"?

A. We specified the biochemical measurements.

The statement on height and weight measurement should come before body mass index calculation.

A. The statement was moved before the body mass index.

Before Statistical Analysis subsection, the authors should provide statements on the sample size determination.

A. Sample size was difficult to calculate based on previous studies. We added the following sentences to the limit of the study.

“Sample size was difficult to draw out from previous studies, as they were carried out in children and patients on dialysis. Our sample of 200 subjects was higher than all the samples in the other studies, thus we believe that our finding may be a reasonable estimate.”

Concerning these statements under “Results”, Vitamin A was determined in 200 ambulatory patients affected by type 2 diabetes. Patients who took vitamin supplements within the six months before the study were excluded. The cut-off levels of vitamin A were ≤ 1.04 and > 2.79 μmol/L for either hypovitaminosis or hypervitaminosis, respectively. These statements cannot be the narratives for the result outcomes. They are better suited for the subsection on methodology.

A. Thanks. We removed the statement because all the information was already present in the methods section.

The authors found that “The sex distribution of HVA was substantially the same, 22.4% women and 32.2% men with a p value of 0.144.” What sampling technique did the authors adopt in arriving at this outcome for the sex distribution?

A. We specified how the patients were enrolled in the methods section.

The authors also suggested that “the levels of VA increased throughout the categories of CKD; G1= 2.26±0.48; G2= 2.40±0.68; G3a= 2.72±0.71; G3b 2.56±0.61 and G4 (only 3 subjects) 3.45±0.79, p for trend <0.001. The increasing level of vitamin A was significant starting from the CKD3a category (post hoc test of Scheffè). These statements of result claimed by the authors in Figure 2 do not exist in the said Figure 2. So it was difficult to verify the outcome.

A. Figure 2 refers to the concentrations of vitamin A according to the presence of either cardiovascular disease or chronic kidney disease. Concentrations of vitamin A in relation to the stages of chronic kidney disease are reported only in the narrative, no figure was drawn from these data.

In the narrative for Table 2, the authors should report the odd ratio, confidence interval and p-values for the association between obesity and HVA.

A. Done

The subsection “Conclusions” should be rendered “Discussions”

A. Done

Under “Conclusions”, The statement “Notably, vitamin A concentration began to increase from stage CKD G3a is not supported by the said "Figure 2 ". Please clarify.

A. As explained the stages of CKD were reported only in the text, no figure was done.

Reviewer #2: I appreciate the opportunity to review this manuscript and contribute to the scholarly process. The study addresses an important yet underexplored aspect of metabolic health, and I look forward to providing constructive feedback.

General Comments:

1. Please add line numbers to facilitate clear and precise communication during the review process.

A. We added lines to the clear revised manuscript.

2. Include a title for the Background or Introduction section after the abstract for better organization.

A. Done

3. Consider structuring the final section as Discussion and Conclusion rather than just Conclusion to ensure a comprehensive synthesis of findings.

A. Done

Specific comments

1. The term "RELAZION" in the title is unclear. Could you clarify its intended meaning? If you meant "relation" or "relationship," I recommend revising the title for clarity and accuracy.

A. Thanks.

2. What are the implications of hypervitaminosis A on diabetes treatment?

A. We believe that our findings can suggest to measure Vitamin A overall before patients start taking multivitamin supplements. Future studies may address the impact of diabetes treatments on vitamin A levels.

3. Does it pose a real concern for diabetes outcomes, or could it potentially influence treatment efficacy?

A. Thanks. All these important questions can be address in dedicated studies.

4. Are there any established guidelines recognizing hypervitaminosis A as a complication, a factor influencing disease progression, or a determinant of treatment outcomes?

A. As far as we know the answer is no.

5. How does this study contribute to the existing body of knowledge, and what is its practical significance for clinicians managing diabetes?

A. At this stage our study is important because it suggest that hypervitaminosis A is present in patients with type 2 diabetes. We think that future studies may focus on the significance of this finding in the treatment of diabetes. Until then we would like to suggest to considered hypervitaminosis A in patients taking vitamin supplements.

6. Are you suggesting that hypervitaminosis A could serve as an indicator of early-stage chronic kidney disease (CKD) in diabetic patients?

A. This is possible. But only a dedicated study can answer to this question.

7. If so, what are the potential mechanisms by which hypervitaminosis A could contribute to or reflect kidney dysfunction in this context?

A. We discuss the hypothetical mechanisms, the definitive answer to this question is not available yet.

8. What are the clinical implications of these findings for early diagnosis and management of CKD in diabetic individuals?

A. The clinical implications should wait the finding of dedicated studies.

Reviewer #3: Title language should be in english - relation and not relazion

A. corrected

Introduction -

Heading is missing

A. corrected

Vitamin A shoild not be abbreviated

A. we replace all the abbreviations.

Retinoic and not retinoid acid

A. corrected

Introduction doesnt emphasize the role of Vitamin A in Type 2 diabetes. Direct implication of Vit A in type 2 diabetes was nto mentioned.

A. Thanks for raising this point. The relations between vitamin A and the risk of diabetes has been studied extensively. We limited our study to investigate the prevalence of hypervitaminosis A in patients with type 2 diabetes.

Research design and method

Inclusion criteria reason out the age range and sex. Was there any grouping done based on age or sex. If not why wasit not done. Did the patients used insulin supplement? Were they included or excluded. Does exclusion criteria inludes any terminal or old age related illness. What were the biochemical blood measurements done. Not reported in the study. Only creatine was done. Renal function test was not carried out. Severity scale was not mentioned in the research design. Grouping based on severity of the diseases was not done.The sentences should not use the word WE or I. Past tense should be used. is to be corrected to was.

"The main clinical and anthropometric characteristics" - where are the anthrometric characteristics

Correlation between Cardiovascular disease (CVD) and Vit A was not done. What are evaluations done with respect cardiovascular.

A. The distribution of hypervitaminosis A was not different according to gender. Most of the patients were treated with oral hypoglycemic agents, 20% of patients were treated with insulin alone or in association. All patients were included in the study. Levels of vitamin A were not different according to therapy (2.26±0.57 diet, 2.45±0.56 hypoglycemic agents, 2.42±0.82 insulin alone or in association, p ANOVA= 0.550). All patients did not have terminal illness. We specified the biochemical blood measurements in the methods section. Renal function was estimated by the formula as reported in the methods section. We correct the sentence and the paste tense. We evaluated the concentrations of vitamin A and the proportion of hypervitaminosis A according to cardiovascular and chronic kidney disease.

Conclusion

Tense correction in sentences. Should be in one tense. "We" should not be used.

Comparison with other studies which were carried out between Vita A and CVD or Vit A and Renal should be added. Revieing past literature with present study should be carried out. The conclusion doesnt provide information on statistical significane or correlation.

A. We review the literature on hypervitaminosis A and found studies only in children or patients in dialysis. The main finding of our study was the correlation with the renal function, that we emphasized in the discussion section.

Review. Significant defiency in the study design and methodology doesnt substanstiate the role of Vit A in CVD and Renal function.

A. We agree. We reinforced this point in the limitation of the study. The cross-sectional design does not allow to draw conclusions on causality but only on associations.

---

## [Decision Letter · Decision Letter 1]

15 Sep 2025

HYPERVITAMINOSIS A IN TYPE 2 DIABETES AND ITS RELAZION WITH RENAL FUNCTION AND CARDIOVASCULAR COMPLICATIONS.

PONE-D-24-58961R1

Dear Dr. Zoppini,

We’re pleased to inform you that your manuscript has been judged scientifically suitable for publication and will be formally accepted for publication once it meets all outstanding technical requirements.

Kind regards,

Tarek Samy Abdelaziz, MD,FRCP

Academic Editor

PLOS ONE

Additional Editor Comments (optional):

Reviewer #2:

Reviewer #3:

Reviewers' comments:

Reviewer's Responses to Questions

**Comments to the Author**

1. If the authors have adequately addressed your comments raised in a previous round of review and you feel that this manuscript is now acceptable for publication, you may indicate that here to bypass the “Comments to the Author” section, enter your conflict of interest statement in the “Confidential to Editor” section, and submit your "Accept" recommendation.

Reviewer #2: All comments have been addressed

Reviewer #3: All comments have been addressed

2. Is the manuscript technically sound, and do the data support the conclusions?

Reviewer #2: No

Reviewer #3: Yes

3. Has the statistical analysis been performed appropriately and rigorously? 

Reviewer #2: Yes

Reviewer #3: Yes

4. Have the authors made all data underlying the findings in their manuscript fully available?

Reviewer #2: Yes

Reviewer #3: Yes

5. Is the manuscript presented in an intelligible fashion and written in standard English?

Reviewer #2: Yes

Reviewer #3: Yes

6. Review Comments to the Author

Reviewer #2: I truly appreciate your response. As a suggestion for future related studies, I strongly believe it is essential to clearly define the significance of the research by identifying specific gaps in the existing literature. Without a well-articulated rationale and clearly stated implications, particularly for patients with type 2 diabetes, such studies may lack direction and could result in a misallocation of valuable resources.

Reviewer #3: The author addressed all queries from the past review..The author had provuded information wherever asked for.

7. PLOS authors have the option to publish the peer review history of their article (what does this mean? ). If published, this will include your full peer review and any attached files.

**Do you want your identity to be public for this peer review?** For information about this choice, including consent withdrawal, please see our Privacy Policy .

Reviewer #2: No

Reviewer #3: No

---

## [Editor Report · Acceptance letter]

PONE-D-24-58961R1

PLOS ONE

Dear Dr. Zoppini,

I'm pleased to inform you that your manuscript has been deemed suitable for publication in PLOS ONE. Congratulations! Your manuscript is now being handed over to our production team.

Kind regards,

on behalf of

Professor Tarek Samy Abdelaziz

Academic Editor

PLOS ONE